# An enhanced therapeutic effect of repetitive transcranial magnetic stimulation combined with antibody treatment in a primate model of spinal cord injury

Hajime Yamanaka[1]*, Yu Takata[1], Hiroshi Nakagawa[1¤], Tomoko Isosaka-Yamanaka[1], Toshihide Yamashita[2], Masahiko Takada[1]

1 Systems Neuroscience Section, Department of Neuroscience, Primate Research Institute, Kyoto University, Inuyama, Aichi, Japan, 2 Department of Molecular Neuroscience, Graduate School of Medicine, Osaka University, Suita, Osaka, Japan

¤ Current address: Department of Molecular Neuroscience, World Premier International Immunology Frontier Research Center, Osaka University, Suita, Osaka, Japan
* yamanaka.hajime.8x@kyoto-u.ac.jp

**Data Availability Statement:** All relevant data are within the paper.

## Abstract

Repetitive transcranial magnetic stimulation (rTMS) targeting the primary motor cortex (MI) is expected to provide a therapeutic impact on spinal cord injury (SCI). On the other hand, treatment with antibody against repulsive guidance molecule-a (RGMa) has been shown to ameliorate motor deficits after SCI in rodents and primates. Facilitating activity of the corticospinal tract (CST) by rTMS following rewiring of CST fibers by anti-RGMa antibody treatment may exert an enhanced effect on motor recovery in a primate model of SCI. To address this issue, we examined whether such a combined therapeutic strategy could contribute to accelerating functional restoration from SCI. In our SCI model, unilateral lesions were made between the C6 and the C7 level. Two macaque monkeys were used for each of the combined therapy and antibody treatment alone, while one monkey was for rTMS alone. The antibody treatment was continuously carried out for four weeks immediately after SCI, and rTMS trials applying a thermoplastic mask and a laser distance meter lasted ten weeks. Behavioral assessment was performed over 14 weeks after SCI to investigate the extent to which motor functions were restored with the antibody treatment and/or rTMS. While rTMS without the preceding antibody treatment produced no discernible sign for functional recovery, a combination of the antibody and rTMS exhibited a greater effect, especially at an early stage of rTMS trials, on restoration of dexterous hand movements. The present results indicate that rTMS combined with anti-RGMa antibody treatment may exert a synergistic effect on motor recovery from SCI.

## Introduction

During the process of functional recovery from central nervous system disorders, effective approaches that strengthen appropriate neural circuits by facilitating their activity is critical.

**Funding:** This work was supported by the Strategic Research Program for Brain Sciences from the Agency for Medical Research and Development (AMED), Japan in the form of a grant (17dm0107117h0002) and Mitsubishi Tanabe Pharma Corporation (Tokyo, Japan) in the form of the anti-RGMa antibody used in the study, funding for the expendables, and a stipend awarded to HY and MT. The funders had no further role in study design, data collection and analysis, decision to publish, or preparation of the manuscript.

**Competing interests:** The authors have read the journal's policy and have the following competing interests: Mitsubishi Tanabe Pharma Corporation provided financial support for the study in the form of funding for the expendables and a stipend awarded to HY and MT, as well as material support in the form of the anti-RGMa antibody used in the study. This does not alter our adherence to PLOS ONE policies on sharing data and materials. There are no other patents, products in development or marketed products associated with this research to declare.

This event can be promoted not only by physical exercise, but also by brain stimulation such as repetitive transcranial magnetic stimulation (rTMS) (for review, see Fitzgerald et al., 2006 [1]). Indeed, rTMS has been shown to induce activation and plasticity of neurons and their circuitry [2–4], and used for clinical application toward therapeutic trials against various diseases, including stroke, multiple sclerosis, Parkinson's disease, depression, and neuropathic pain [5, 6]. Currently, rTMS therapy mainly targeting the primary motor cortex (MI) is expected to improve motor impairments by activating neural circuits related to limb movements [7, 8]. Some studies in human patients have, in fact, reported that rTMS trials over the MI exert a certain therapeutic impact on spinal cord injury (SCI) [9, 10].

On the other hand, repulsive guidance molecule-a (RGMa) is known as a functional molecule that primarily inhibits axonal growth. Recent emphasis has been placed on diverse effects of its neutralizing antibody in both basic and clinical research works [11]. The antibody against RGMa has increasingly been recognized as a promising agent for treatment of a wide variety of diseases and dysfunctions, including malignant tumors, autoimmune and allergic diseases, stroke, and multiple sclerosis [12–14]. Meanwhile, it has been demonstrated in rodents that administration of the anti-RGMa antibody ameliorates motor impairments caused by SCI [15, 16]. We have also provided evidence that a similar therapeutic effect of antibody treatment is obtained in a primate model of SCI [17]. Here we hypothesized that rewiring of corticospinal tract (CST) fibers through anti-RGMa antibody treatment, followed by facilitation of CST activity through rTMS targeting the MI, might contribute to accelerating motor recovery from SCI. To test this hypothesis, we analyzed a possible enhanced effect of such a combined therapeutic strategy on functional restoration in the SCI model using macaque monkeys. To archive constant and effective magnetic stimulation over the MI of awake animals, we have developed a novel system in which a thermoplastic mask and a laser distance meter were applied (for the usefulness of thermoplastic mask in rTMS, see Drucker et al., 2015 [18]).

## Materials and methods

### Animals and experimental design

Seven adult male Japanese monkeys (*Macaca fuscata*; 5.4–7.1 kg body weight) were used. The monkeys, all of which were subjected to SCI, were divided into four groups: two monkeys (M, Z) with anti-RGMa antibody treatment alone; one monkey (G) with rTMS alone; two monkeys (A, C) with both the antibody treatment and rTMS; and two monkeys (I, Y) without either the antibody treatment or rTMS. The experimental design is summarized in Fig 1A. In each of the seven monkeys tested here, a modified Brinkman board test was trained for two-to-three months prior to SCI, as described elsewhere [17]. Following SCI, behavioral assessment was performed over 14 weeks to examine the extent to which motor functions were restored. At the same timing as the SCI surgery, antibody treatment was carried out in four of the seven monkeys. The anti-RGMa antibody was continuously delivered to the lesioned site for four weeks via an osmotic infusion pump. Shortly after the completion of antibody administration (four to five weeks after SCI), two of the four monkeys underwent trials of rTMS over the MI, especially its forelimb region, three to five times a week for a total of approximately ten weeks. To identify CST fibers derived from the contralesional MI, biotinylated dextran amine (BDA) was injected thereinto before sacrifice. The experimental protocols were approved by the Animal Welfare and Animal Care Committee of the Primate Research Institute, Kyoto University (Permission No. 2016–053). All experiments were conducted according to the Guidelines for Care and Use of Nonhuman Primates (Ver. 3, 2010) prepared by the Primate Research Institute. All monkeys used in this study, weighing less than 10 kg, were kept in indoor individual cages (0.89 m×0.63 m×0.82 m) on a 12-h on/12-h off lighting schedule. In each cage, the

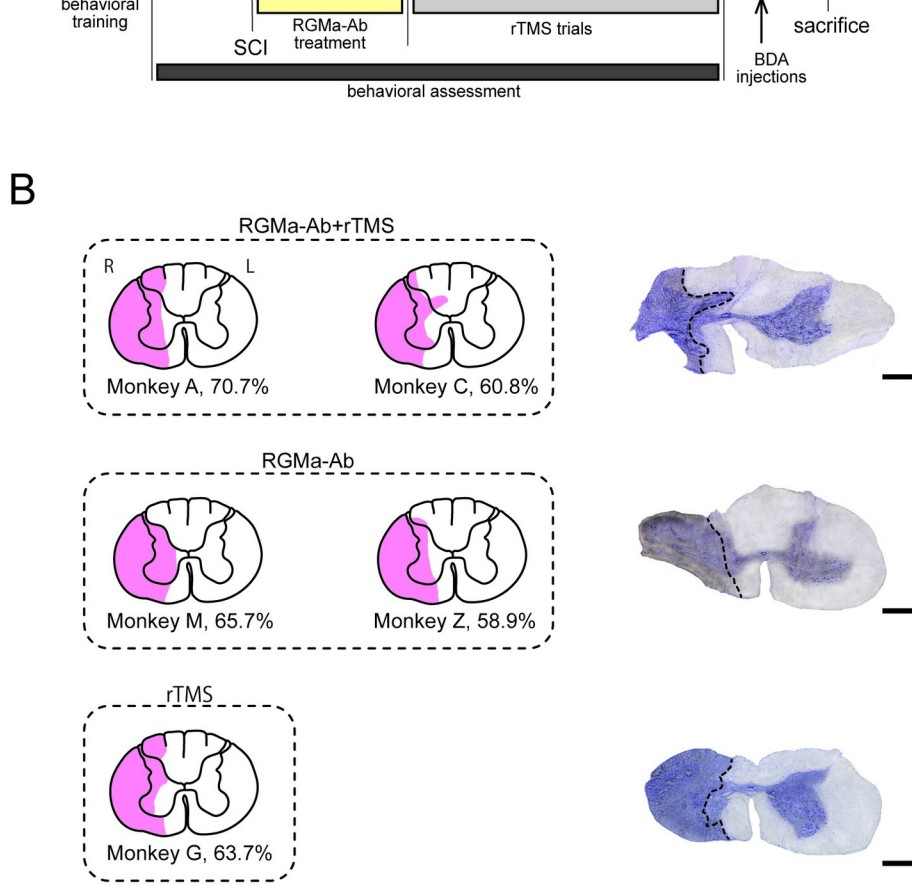

**Fig 1. Experimental design and extent of SCI lesions.** (A) Experimental design. According to a modified Brinkman board test, dexterous motor behavior was assessed over a total of 16 weeks (wks) before and after SCI (2 and 14 weeks, respectively). Treatment with anti-RGMa antibody (RGMa-Ab) was started concurrently with the SCI surgery and continued for four weeks. Following the antibody treatment, rTMS trials over the forelimb region of the MI were performed for a total of approximately 10 weeks. Then, BDA injections into the contralesional MI were made about eight weeks before sacrifice. (B) Extent of SCI lesions (in pink) overlaid on a template transverse section. Lesions were made unilaterally at the border between the C6 and the C7 level to involve largely the lateral sector of the cervical cord. Monkeys A and C with both the antibody treatment and rTMS (RGMa-Ab+rTMS), monkeys M and Z with the antibody treatment alone (RGMa-Ab), monkey G with rTMS alone (rTMS), and monkeys I and Y without either the antibody treatment or rTMS (Control SCI). Each percentage represents the ratio of the lesioned area to the total area of a hemi-transverse section. Note that there is no marked difference in the extent of SCI lesions (58.9–70.7%) across the seven monkeys. L, left; R, right. On the right side, photomicrographs of representative sections Nissl-stained with Cresyl violet taken from monkeys C, Z, G, and Y. Scale bars, 1 mm.

monkeys were given *ad libitum* access to food and water, and a chain-hung wood block as a toy. The monkeys were monitored closely, and animal welfare was assessed on a daily basis and, if necessary, several times a day. This includes veterinary examinations to make sure animals are not suffering. If animals experience pain, they receive pain medications. If pain cannot be relieved, or if veterinary examinations reveal signs of suffering that cannot be relieved by analgesics, antiemetics, or antibiotic therapy, animals are euthanized.

## Behavioral assessment

A modified Brinkman board test was employed to assess quantitatively the extent of motor recovery (i.e., manual dexterity) from SCI, as described elsewhere [17]. Briefly, the Brinkman board (100 mm x 200 mm) was placed on a monkey chair. A total of 50 slots (15 mm long x 8 mm wide x 6 mm deep each), consisting of 25 vertical and 25 horizontal slots, were randomly located on the board. Each slot was filled with a fruit-flavored pellet (5 mm in diameter, 94 mg; BioServe, NJ, USA). The monkeys were trained to take out as many pellets as possible within 30 sec (one session) and forced to perform consecutive five sessions a day. The behavioral assessment based on the modified Brinkman board test was done twice a week (three to four days apart). When the monkey took out a pellet from a vertical or horizontal slot to convey it to the mouth successfully, each performance was defined as a vertical- or horizontal-slot task, respectively. Such motor performance was evaluated as the number of pellets collected per session (through both the vertical and the horizontal slots, the vertical slots only, or the horizontal slots only).

## Anti-RGMa antibody treatment

A humanized monoclonal antibody against human RGMa was used in this study. The antibody was kindly donated by Mitsubishi Tanabe Pharma Corporation (Tokyo, Japan). The amino-acid sequence in a putaive epitope region of RGMa for humans was nearly identical (98%) to that for macaque monkeys. As previously described [17], the anti-RGMa antibody was continuously delivered (50 μg/kg/day) to the periphery of the lesioned site, through an osmotic infusion pump (Alzet 2ML4, Durect Co., CA, USA) which was installed under the skin of the back, over four weeks immediately after SCI. To fix properly a catheter (11 cm long) attached to the osmotic pump, the top of the catheter was placed 5 mm above the lesioned site under the dura mater (intrathecal administration), and the catheter was connected to the dura mater and surrounding muscles with surgical suture to keep the position during delivery of the antibody. Four weeks later, the osmotic pump was removed from the back and then checked to ensure that the whole amount of the antibody was certainly delivered up.

## SCI surgery

The SCI surgery was carried out in almost exactly the same manner as previously describe [17, 19]. Briefly, the monkeys were sedated with a combination of ketamine hydrochloride (6.7 mg/kg, i.m.), xylazine hydrochloride (1.3 mg/kg, i.m.), and atropine (0.05 mg/kg, i.m.), and then underwent sevoflurane anesthesia (1–3%) following tracheal intubation. Laminectomy of the C3 to C7 segments was done, and the dura mater was cut unilaterally. After identification of the dorsal roots at the C6 and C7 levels, the border between the C6 and the C7 segment was lesioned with a surgical blade (No. 11) and a special needle (27 G), to infringe upon the dorsolateral funiculus where the CST travels. Part of the gray matter was kept intact to retain a route of sprouting CST fibers. After SCI, the dura mater was sutured, and then the

skin and axial muscles were sutured. This procedure was done as consistently as possible among the animals.

### TMS trials

Magnetic stimulation was given to the MI in the hemisphere contralateral to SCI through an air-cooled, 70-mm in diameter, figure-of-eight coil connected to a Magstim Rapid$^2$ Stimulator (Magstim Co., Carmarthenshire, Wales, UK; see https://www.magstim.com/product/rapid-family/ for details) which allows repetitive, biphasic, and cortical/peripheral stimulation. Previous studies [1, 20, 21] have reported that rTMS at the frequency of 10 Hz or higher facilitates neural activity of the stimulated site, and this effect lasts at least several days. Furthermore, a research group of Dr. Ken-ichiro Tsutsui at Tohoku University has recently conducted a systematic experiment using monkeys to evaluate facilitatory and inhibitory effects of rTMS at different frequencies through electrocorticography and motor-evoked potential recordings. They have shown that intermittent rTMS at a 10 Hz or higher frequency has a long-lasting facilitatory effect on neural activity (personal communications). Accordingly, we set stimulation parameters as 15–20 Hz frequency, 3-sec train duration, and 17-sec inter-train interval. Based on the previous reports [1, 20, 21] and our personal communications with Dr. Tsutsui at Tohoku University, the total duration of stimulation sessions (40–46 trains per session) and the total number of pulses (2025–2400 pulses over 15 min) were set to have an enough margin to avoid unexpected overactivation of neural circuits. Since side effects of rTMS, such as epilepsy and headache, may occur, we carefully determined the stimulation parameters while observing individual physical conditions. In fact, the frequency and train number were set as milder at the onset of rTMS, changed every week during the first three weeks, and then kept constant: (1) 15 Hz, 45 trains, 2025 pulses for the first week; (2) 17 Hz, 46 trains, 2346 pulses for the second week; and (3) 20 Hz, 40 trains, 2400 pulses for third week. After the fourth week, the parameters were fixed in the same conditions as for the third week until the final week. Sessions of rTMS were performed once a day, three to five days a week and continued for ten weeks. The total number of sessions was 44–50 for a single monkey. To apply rTMS to the MI of an awake monkey, a threshold was defined as the minimum stimulation intensity to induce simple forelimb movement which was identified by visual inspection. The stimulation intensity was 100–105% of such a threshold, ranging from 60 to 71% of the maximum stimulator output. In cooperation with late Dr. Yasushi Kobayashi at Osaka University, a thermoplastic mask (MTAPUR, CIVCO Medical Solutions, IA, USA) was utilized to immobilize the monkey's head only during the rTMS sessions, and laser distance meters (GLM 50C, Bosch, Gerlingen, Germany) were used to measure the three-dimensional coordinates of the figure-of-eight coil.

### Anterograde tract-tracing of CST fibers

To examine the intraspinal distribution of CST fibers arising from the contralesional MI, anterograde tract-tracing with BDA was performed as previously described [17, 19, 22]. Briefly, a 10% solution of BDA (10,000 MW; Invitrogen, CA, USA) in 0.1 M phosphate-buffered saline (PBS; pH 7.4) was injected into the MI after the behavioral assessment. Under general anesthesia with sevoflurane (1–3%), the monkeys underwent craniotomy to expose the central sulcus. Multiple injections of BDA (150 nl x 84 sites) were made extensively over the forelimb region of the MI under visual inspection through a 5-μL Hamilton microsyringe. After the BDA injections, the dural defect was covered with a gelfilm (Pfizer, MI, USA), and the craniotomy site was covered with a piece of cranium and fixed with acrylic resin.

## Histological and statistical analyses

The monkeys were allowed to survive for about eight weeks following the BDA injections and sacrificed by perfusion-fixation. After sedation with ketamine (8 mg/kg, i.m.) and medetomidine (0.04 mg/kg, i.m.), the monkeys were anesthetized deeply with an overdose of sodium pentobarbital (50 mg/kg, i.v.) and perfused transcardially with 0.1 M PBS, followed by 10% formalin in 0.1 M phosphate buffer. The brain and spinal cord were immediately removed, postfixed in the same fixative overnight, and then saturated with 30% sucrose. The cervical enlargement of the spinal cord was cut serially into 40-mm-thick transverse sections on a freezing microtome. In each case, SCI lesions were demarcated in representative sections Nissl-stained with 1% Cresyl violet and overlaid on a template section at the C6/C7 level. The extent of SCI lesions was measured using ImageJ software, and for identification of significant outliers in the lesion extent, Grubb's outlier test was utilized with the significance level of 0.05 (Fig 1B).

To examine the infiltration of microglial cells, every 10th section was processed for immunohistochemical staining for ionized calcium binding adapter molecule 1 (Iba1). Floating sections were initially treated with 0.3% $H_2O_2$ for blocking the endogenous peroxidase. After immersion with 1% skim milk for 60 min, the sections were incubated with rabbit anti-Iba1 antibody (1:2,000 dilution; Wako, Osaka, Japan) in 0.1 M PBS containing 2% normal donkey serum, and 0.1% Triton X-100 for 120 min at room temperature, followed by 2–3 overnights at 4˚C. Subsequently, the sections were incubated with biotinylated donkey anti-rabbit IgG antibody (1:1,000 dilution; Jackson Immuno Research laboratories, PA, USA) in 0.1 M PBS containing 1% normal donkey serum for 2 hr at room temperature, and then with the avidin-biotin-peroxidase complex kit (ABC Elite; 1:200 dilution; Vector laboratories, CA, USA) for 90 min at room temperature. Finally, the sections were reacted in 0.05 M Tris-HCl buffer (pH 7.6) containing 0.01% diaminobenzidine tetrahydrochloride (DAB), 0.02% nickel chloride,and 0.0006% $H_2O_2$, and then counterstained with 0.5% Neutral red. On the other hand, to visualize CST fibers anterogradely labeled with BDA, floating sections were initially treated with 0.3% $H_2O_2$ and then with the ABC kit (1:200 dilution) for 90 min at room temperature, followed by one overnight at 4˚C. Subsequently, the sections were reacted in 0.05 M Tris-HCl buffer containing 0.04% DAB, 0.04% nickel chloride, and 0.0001% $H_2O_2$, and then counterstained with Neutral red. All immunohistochemical and histochemical images were acquired with a light microscope (BZ-X800, Keyence, Osaka, Japan).

## Results

### Development of rTMS system

First of all, we developed an rTMS system to confer magnetic stimulation to the MI of awake monkeys. As shown in Fig 2, our rTMS system consisted of (1) a magnetic stimulator with a figure-of-eight coil, (2) a coil fixation device attached to a monkey chair fixation frame, (3) a monkey chair and a thermoplastic mask, and (4) laser distance meters and their fixation device. To conduct magnetic stimulation constantly and effectively, two major modifications were made in the present system. One was the application of a thermoplastic mask, instead of a head fixation apparatus ordinarily used for electrophysiological recording, to fix the monkey's head and secure a sufficient space for a huge figure-of-eight coil relative to the head. The mask was shaped to fit the monkey's head for restraining any head movement and contacting the coil close to the scalp (Fig 2A–2C). Another modification was to introduce a three-dimensional coil positioning system. The position of the figure-of-eight coil was determined by the aid of three laser distance meters that could measure the spatial coordinates of the coil. In this

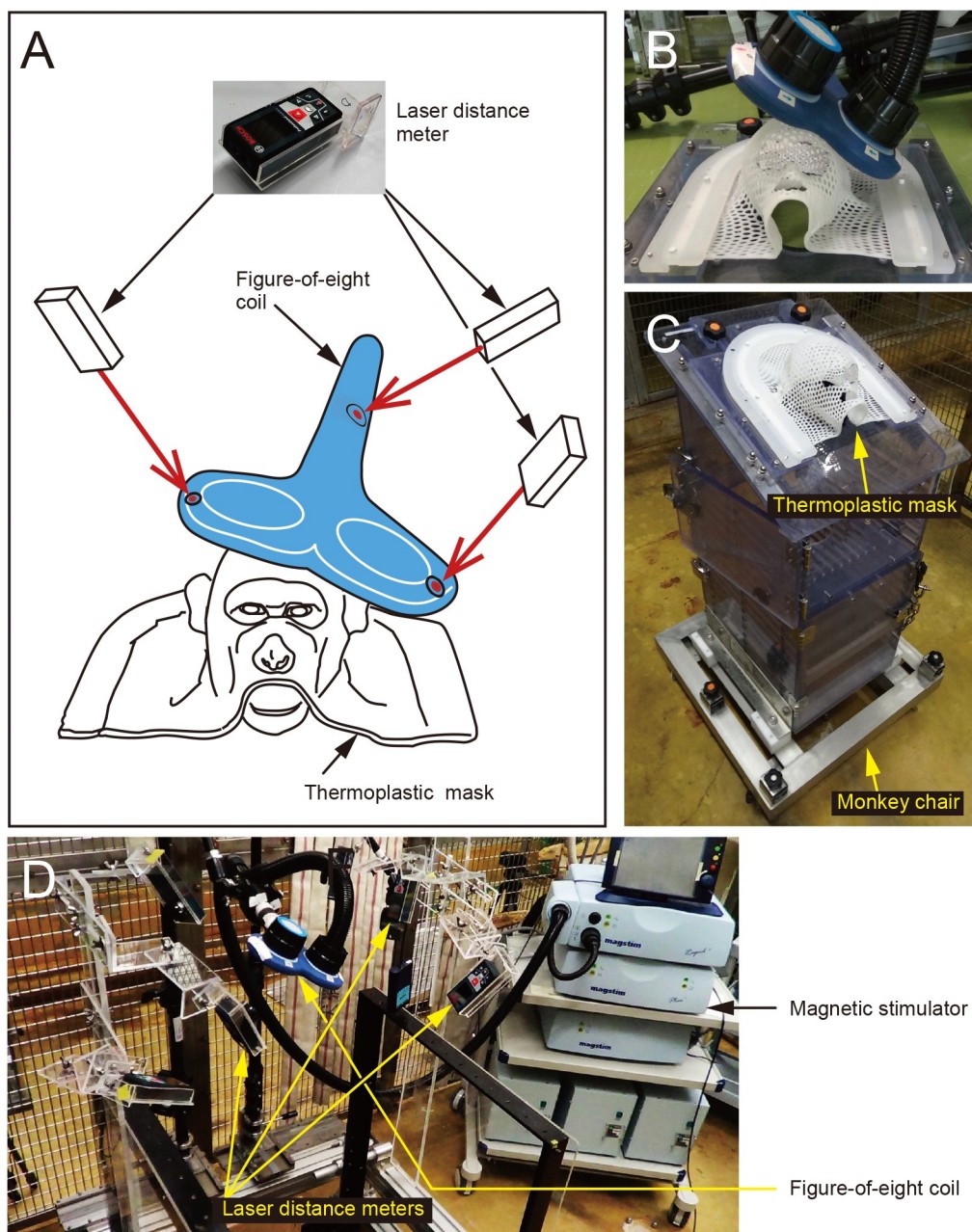

**Fig 2. Setup of the newly developed rTMS system.** (A) Schematic diagram of the present system for achieving constant and effective stimulation targeting the MI of an awake monkey. (B) A thermoplastic mask shaped to fit the monkey's head is utilized to fix the monkey's head, secure a sufficient space for a figure-of-eight coil, and contact the coil close to the scalp. (C) The thermoplastic mask is firmly screwed to the upper part of a monkey chair for restraining any head movement. (D) Three laser distance meters are positioned to measure accurate three-dimensional coordinates of the figure-of-eight coil. For details, see "Development of rTMS system" of the Results section.

way, the coil was placed in the same position before and after SCI to perform rTMS targeting an appropriate cortical area, i.e., the forelimb region of the MI, with good reproducibility (Fig 2A, 2B and 2D).

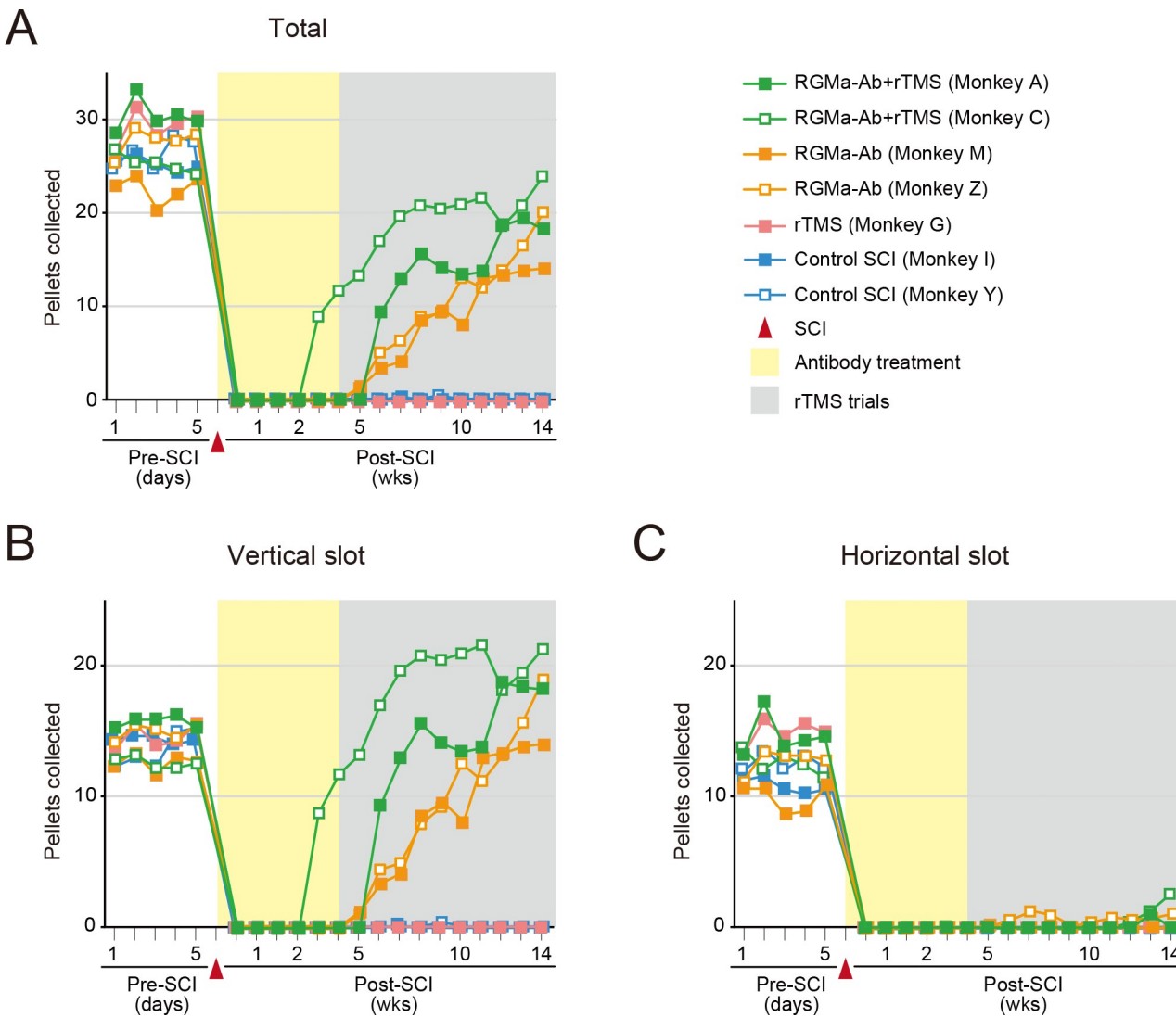

**Fig 3. Results of behavioral assessment based on the modified Brinkman board test in the seven monkeys tested.** (A) Total number of pellets collected through both the vertical and the horizontal slots. (B) Number of pellets collected through the vertical slots. (C) Number of pellets collected through the horizontal slots. Note that rTMS combined with the antibody treatment appears to yield a more rapid improvement in the motor performance requiring manual dexterity, compared with the antibody treatment alone.

## SCI model

In our SCI model, lesions were made unilaterally at the border between the C6 and the C7 level to involve largely the lateral sector of the cervical cord. The lesioned area was expressed as the ratio to the total area of a hemi-transverse section, and there was no marked difference in the extent of SCI lesions (58.9–70.7%) across the seven monkeys (Fig 1B; Grubb's outlier test, $p > 0.05$). Therefore, all these monkeys completely lost their capability in forelimb movement immediately SCI (see after Fig 3). In each case, CST fibers derived from the contralesional MI were identified by anterograde tract-tracing with BDA. We found that such crossed CST fibers were virtually removed below the SCI lesion site in the non-treated cases, while a number of CST fibers extended across the lesion site into laminae VII–IX in the treated cases with

antibody infusion and/or rTMS (S1A and S1B Fig). In addition, the infiltration of microglial cells around the SCI lesions was confirmed with Iba1 immunohistochemistry. No marked microglial accumulations were observed, although Iba1-positive cells were scattered more frequently in the non-treated cases than in the treated cases (S1C and S1D Fig).

## Effects of rTMS combined with antibody treatment

Using the modified Brinkman board test as an index, we analyzed the effects of antibody treatment and/or rTMS on the recovery of motor functions, i.e., manual dexterity, after SCI. Throughout the entire period of behavioral assessment (14 weeks), virtually no functional recovery from SCI was seen in two control monkeys without either the antibody treatment or rTMS (Fig 3A). Likewise, no sign indicative of motor recovery was found at all in one monkey with rTMS only (Fig 3A). Both of two monkeys with antibody treatment only, on the other hand, displayed progressive recovery of dexterous motor functions immediately after the completion of antibody administration. The motor performance was prominently improved in a vertical-slot task and returned to the pre-SCI level around 10 weeks following the onset of antibody administration (Fig 3A and 3B). A combination of antibody treatment and rTMS targeting the MI resulted in a similar and more rapid improvement in skilled motor behavior, compared with the antibody treatment alone. In one of the two monkeys, the motor performance in the vertical-slot task was restored more quickly to reach as high as the pre-SCI level within a few weeks following the onset of rTMS (i.e., a few weeks earlier than in the case of antibody treatment alone) (Fig 3A and 3B). Another monkey also seemed to exhibit a marked recovery of motor functions from SCI, though the start of recovery preceded the onset of rTMS as well as the completion of antibody administration (Fig 3A and 3B). None of the monkeys had a discernible improvement in the motor performance in a horizontal-slot task which requires more dexterous hand movements (Fig 3C). The monkeys who underwent rTMS combined with the antibody treatment tended to take out more pellets from vertical rather than horizontal slots during each session (Fig 3B). Thus, the addition of rTMS to the antibody treatment enhanced the rate of behavioral improvement at an early stage of rTMS trials. However, the final level of motor recovery at the 14th week post-SCI in the two animals receiving rTMS was not largely different from that in the two animals not receiving rTMS (Fig 3A and 3B).

## Discussion

Although meta-analysis of the therapeutic effects of rTMS was generally performed in three key aspects of motor/sensory functions, spasticity, and neuropathic pain [23], the outcome especially concerning the motor/sensory functions was not always consistent in human patients (see Belci et al., 2004; Benito et al., 2012 for success [9, 10], but see Kuppuswamy et al., 2011 for failure [24]). Our rTMS trials targeting the MI, on the other hand, failed to improve dexterous motor deficits in the present monkey model of SCI. These contradictory results might depend on varying trial conditions, such as SCI severity and stage, treatment timing, and rTMS parameters.

The effectiveness of antibody therapy to neutralize RGMa, an inhibitor of axonal growth, has also been explored in both rodent and primate models of SCI [15–17, 25]. In our prior study [17], the impact of anti-RGMa antibody treatment on functional recovery was investigated in macaque monkeys whose manual dexterity was impaired after SCI at the cervical enlargement level. It was clearly demonstrated in a series of experiments that the antibody treatment exerted a remarkable therapeutic effect, and that fibers of the CST derived from the MI contralateral to SCI travelled beyond the lesioned site following the antibody treatment. We have further reported that many of the CST fibers sprouting from the contralesional MI

prominently extend into the motoneuron pool with the recovery from impaired manual dexterity [19].

The present data obtained in our monkey SCI model indicate that rTMS over the MI combined with anti-RGMa antibody treatment may yield a more rapid improvement, than the antibody treatment alone, in the motor performance requiring manual dexterity, though one monkey started to recover a few weeks preceding rTMS trials. A similar quick effect of the antibody treatment on functional restoration has been observed in another primate model of SCI using rhesus monkeys (*Macaca mulatta*) [17]. This can be accounted for by postulating the individual difference in the sensitivity of the anti-RGMa antibody. Moreover, although rTMS following the antibody treatment exhibited a greater therapeutic effect, especially at an early stage of rTMS trials, the final levels of motor recovery at the 14th week post-SCI were similar in the monkeys with and without rTMS. This suggests that the prolonged use of rTMS may not be so effective as one expect. The same therapeutic approach using a combination of anti-RGMa antibody treatment and rTMS has been shown to promote motor recovery in a mouse SCI model [26]. In their study, rTMS trials targeting the motor cortex subsequent to the antibody treatment successfully exerted a synergistic effect in comparison with the antibody treatment alone, while rTMS performed concurrently with the antibody treatment did not.

Our anterograde tract-tracing with BDA revealed that the CST was completely dissected out in the present SCI model, and, therefore, virtually no CST fibers derived from the contralesional MI were observed below the SCI lesion site. Consistent with the findings in our prior works [17, 19], a number of CST fibers were seen to extend across the lesion site to laminae VII–IX in the cases with antibody treatment and/or rTMS trials, indicating an anatomical sign of functional recovery. Moreover, Iba1 immunohistochemistry was carried out to assess the extent of SCI lesions, since microglial infiltration as an inflammatory response occurs around the lesion site, especially in the acute phase [17]. In our analysis, however, accumulations of microglial cells were less drastically found than expected even in the cases without either antibody treatment or rTMS trials, given that inflammatory/immune responses seemed to be rather attenuated by the time of observation at more than six months after the SCI onset.

Overall, the anti-RGMa antibody treatment followed by rTMS over the MI could provide an enhanced effect on motor recovery from SCI. The sequence of this combined therapeutic strategy seems critical for accelerating functional restoration: initially inducing rewiring of CST fibers at the spinal axon-terminal site by the antibody treatment and, then, reinforcing the corticospinal connectivity by facilitating the CST activity at the cortical cell-body site through rTMS trials. Further investigations are called for to elucidate the exact mechanism underlying such neuroplastic events, especially the rTMS effect. It should be stated here, however, that although additional experimental cases must be needed particularly to verify the validity of a synergistic effect of our combined therapy, we unfortunately could not do so, because the institute would not permit us to sacrifice more animals for an invasive SCI model based on the 3Rs principle. In conclusion, the present study implies that rTMS therapy may play a potential role in promoting functional recovery from SCI when the antibody treatment precedes it for rewiring of CST fibers. Also, as we stated the relevance to rTMS in "Development of rTMS system" of the Results section, our developed system using a thermoplastic mask and a laser distance meter might facilitate rTMS-related research works in awake monkeys.

## Supporting information

**S1 Fig. Identification of CST fibers and infiltration of microglial cells.** (A, B) CST fibers anterogradely labeled with BDA injected into the contralesional MI at the spinal levels above (C5) and below (C8/T1) the SCI lesion site in monkeys I (A; Control SCI) and C (B; RGMa-Ab+rTMS). Right panels are higher magnifications of the dorsolateral funiculus (dotted squares), and panels at the bottom are higher magnifications of lamina VII (rectangles) in the sections at the C8/T1 levels. (C, D) Microglial infiltration, as confirmed with Iba1 immunostaining, just below the SCI lesion site in monkeys I (C; Control SCI) and C (D; RGMa-Ab +rTMS). Lower panels are taken from the rectangles in the whole sections (upper panels). All sections were counterstained with Neutral red. Scale bars, 1 mm.
(TIF)

## Acknowledgments

We thank T Koitabashi for technical assistance in animal experiments and data collections.

## Author Contributions

**Conceptualization:** Toshihide Yamashita, Masahiko Takada.

**Data curation:** Hajime Yamanaka, Masahiko Takada.

**Formal analysis:** Hajime Yamanaka.

**Funding acquisition:** Toshihide Yamashita, Masahiko Takada.

**Investigation:** Hajime Yamanaka, Yu Takata, Tomoko Isosaka-Yamanaka.

**Methodology:** Hajime Yamanaka, Hiroshi Nakagawa.

**Project administration:** Masahiko Takada.

**Supervision:** Hajime Yamanaka, Masahiko Takada.

**Validation:** Hajime Yamanaka, Masahiko Takada.

**Visualization:** Hajime Yamanaka, Masahiko Takada.

**Writing – original draft:** Hajime Yamanaka, Masahiko Takada.

**Writing – review & editing:** Hajime Yamanaka, Hiroshi Nakagawa, Toshihide Yamashita, Masahiko Takada.

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
