## [Decision Letter · Decision Letter 0]

17 Dec 2020

PONE-D-20-28971

An enhanced therapeutic effect of repetitive transcranial magnetic stimulation combined with antibody treatment in a primate model of spinal cord injury

PLOS ONE

Dear Dr. Yamanaka,

Thank you for submitting your manuscript to PLOS ONE. After careful consideration, we feel that it has merit but does not fully meet PLOS ONE’s publication criteria as it currently stands. Therefore, we invite you to submit a revised version of the manuscript that addresses the points raised during the review process.   Please carry out the necessary new experiments and data expansion as per the reviewer's requests and re-write the manuscript according to the suggested improvements and new data.  Please perform point-to-point changes in the the manuscript according to the reviewer's suggestions and queries and provide a detailed "Reply to reviewer" letter in which you answer all the raised points and explain all the changes you have made in the revision.

We look forward to receiving your revised manuscript.

Kind regards,

Antal Nógrádi, M.D., Ph.D., D.Sc.

Academic Editor

PLOS ONE

Journal Requirements:

2. In order to comply with PLOS ONE's guidelines for non-human primate experiments (http://journals.plos.org/plosone/s/submission-guidelines#loc-non-human-primates), please provide additional details regarding housing conditions, feeding regimens, environmental enrichment, and all relevant steps taken to alleviate suffering (analgesia, details about humane endpoints and their application, etc.). Also indicate how often animal care staff monitored the health and well-being of the animals and the criteria used to make such assessments.

Reviewers' comments:

Reviewer's Responses to Questions

**Comments to the Author**

1. Is the manuscript technically sound, and do the data support the conclusions?

Reviewer #1: Yes

2. Has the statistical analysis been performed appropriately and rigorously? 

Reviewer #1: Yes

3. Have the authors made all data underlying the findings in their manuscript fully available?

Reviewer #1: No

4. Is the manuscript presented in an intelligible fashion and written in standard English?

Reviewer #1: Yes

5. Review Comments to the Author

Reviewer #1: This is an interesting paper focusing on a combined therapeutic approach to treat spinal cord injury (SCI). The authors examined whether a combined application of repetitive transcranial magnetic stimulation (rTMS) and repulsive guidance molecule-a (RGMa) is able to enhance the functional restoration of SCI in Macaca fuscata. A unilateral lesion was performed in the lower cervical spinal cord in adult male monkeys. Behavioral assessment was carried out over 14 weeks after SCI. Although the authors made serious efforts to prove the proposed hypothesis, the manuscript suffers from several flaws.

1) The cervical enlargement of the spinal cord was cut serially into 40-µm thick cross sections for Nissl staining. Surprisingly, the authors did not present any images from stained specimens. It is incomprehensible why no parallel sections were cut from such valuable samples. It would have been possible to examine glia activation and other changes by using immunohistochemistry, etc.

2) The authors repeatedly mentioned the rewiring/sprouting of corticospinal tract (CST). To detect the integrity of the main CST should have been assessed by using immunostaining for Protein kinase C gamma (PKCγ), which specifically labels the main CST. It would have been possible to examine the CST at any distances rostrally and caudally from the injury center.

Although the functional results are promising, it is difficult to judge the outcome of the treatment strategy in the absence of morphological results. It is recommended, that the authors revise their data and improve the morphological analysis to an extent that these could be used for correlation with the functional data. Without these changes the MS cannot be improved to an extent that makes it acceptable for scientific publication.

6. PLOS authors have the option to publish the peer review history of their article (what does this mean?). If published, this will include your full peer review and any attached files.

Reviewer #1: No

---

## [Author Response · Author response to Decision Letter 0]

1 Feb 2021

We are grateful to the Reviewer for providing us with precious comments. Here are point-by-point responses to the comments raised by the Reviewer.

Responses to the comments raised by the Reviewer 

Comment 1

The cervical enlargement of the spinal cord was cut serially into 40-µm thick cross sections for Nissl staining. Surprisingly, the authors did not present any images from stained specimens. It is incomprehensible why no parallel sections were cut from such valuable samples. It would have been possible to examine glia activation and other changes by using immunohistochemistry, etc.

Response to Comment 1

According to this comment, we have added photomicrographs of representative Nissl-stained sections (taken from four cases) in Figure 1B to show actual images of SCI lesions. As described in the Materials and methods section (Histological and statistical analyses; page 10, lines 209-211), SCI lesions were demarcated in the Nissl-stained sections and overlaid on a template section at the C6/C7 level, considering some deformation and tissue damage due to scar formation after SCI lesions.

Comment 2

The authors repeatedly mentioned the rewiring/sprouting of corticospinal tract (CST). To detect the integrity of the main CST should have been assessed by using immunostaining for Protein kinase C gamma (PKCγ), which specifically labels the main CST. It would have been possible to examine the CST at any distances rostrally and caudally from the injury center.

Response to Comment 2

According to this comment, we have added some data on PKC� immunohistochemistry as S1 Fig (A, B) to identify the main CST at the levels rostral (C5) and caudal (C8) to SCI lesions. Moreover, also in relation to Comment 1, we have provided some data on Iba1 immunohistochemistry as S1 Fig (C, D) to examine the infiltration of microglial cells around SCI lesions at the C6/C7 level.

---

## [Decision Letter · Decision Letter 1]

4 Mar 2021

PONE-D-20-28971R1

An enhanced therapeutic effect of repetitive transcranial magnetic stimulation combined with antibody treatment in a primate model of spinal cord injury

PLOS ONE

Dear Dr. Yamanaka,

Thank you for submitting your manuscript to PLOS ONE. After careful consideration, we feel that it has merit but does not fully meet PLOS ONE’s publication criteria as it currently stands. Therefore, we invite you to submit a revised version of the manuscript that addresses the points raised during the review process. Please perform the required changes suggested by the reviewer and explain in the cover letter what changes have been done in the MS.  

We look forward to receiving your revised manuscript.

Kind regards,

Antal Nógrádi, M.D., Ph.D., D.Sc.

Academic Editor

PLOS ONE

Journal Requirements:

Reviewers' comments:

Reviewer's Responses to Questions

**Comments to the Author**

1. If the authors have adequately addressed your comments raised in a previous round of review and you feel that this manuscript is now acceptable for publication, you may indicate that here to bypass the “Comments to the Author” section, enter your conflict of interest statement in the “Confidential to Editor” section, and submit your "Accept" recommendation.

Reviewer #1: All comments have been addressed

2. Is the manuscript technically sound, and do the data support the conclusions?

Reviewer #1: Yes

3. Has the statistical analysis been performed appropriately and rigorously? 

Reviewer #1: Yes

4. Have the authors made all data underlying the findings in their manuscript fully available?

Reviewer #1: Yes

5. Is the manuscript presented in an intelligible fashion and written in standard English?

Reviewer #1: Yes

6. Review Comments to the Author

Reviewer #1: It is evident from the revised manuscript, that the authors made serious efforts to improve the manuscript and give a reply to all comments. Indeed, the quality of the manuscript has improved a lot, but there are still few points which need clarification.

Figure 1. Images of representative Nissl-stained sections taken from monkeys show weak tissue integrity at the intact left side. It appears at first glance, as if the tissue underwent freezing damages, but is not clear what and how happened. It appears that the injured side contains a relatively intact scar tissue. Please explain and correct these images.

Suppl. Fig. 1A and B. Why do these images overlap? Please indicate the injured and intact side. Please show the CST at higher magnification (insert high power images as inset, or similar) to visualize better the difference between the groups. The PKCgamma signal is stronger on the injured side than on the intact one. Please explain and amend/replace the figure.

Suppl. Fig 1C and D. Please insert an overview of spinal cords in lower magnification and indicate the source of higher magnification images (with boxes of sampling).

Furthermore, authors need to indicate the staining in the Fig.1 and Suppl. Fig.1., too.

In results section the glia and CST results should be better explored. At present it is just one sentence. Please explicitly discuss them in the discussion section.

7. PLOS authors have the option to publish the peer review history of their article (what does this mean?). If published, this will include your full peer review and any attached files.

Reviewer #1: No

---

## [Author Response · Author response to Decision Letter 1]

16 Apr 2021

We are grateful to the Reviewer for critical reading of our revised manuscript and providing us with further insightful comments. Here are point-by-point responses to the Reviewer’s comments.

First of all, we apologize for our misleading presentation of PKC� immunohistochemical data in Supplementary Figure 1. For analyzing the extent of SCI lesions through identification of CST fibers, the anterograde tracer, biotinylated dextran amine (BDA), was in fact injected into the primary motor cortex contralateral to the SCI lesion site in all monkeys, Nevertheless, we totally failed to take this into consideration when the original manuscript was prepared. To visualize CST fibers anterogradely labeled with BDA, the use of the avidin-biotin-peroxidase complex is indispensable. This obviously makes it untenable to evaluate immunohistochemical data to be obtained with the same method, especially at the spinal level above the SCI lesion site. In this revised version, we therefore would like to replace the data on PKC� immunostaining with those on BDA anterograde tract-tracing (see Supplementary Figure 1A and B), if the Reviewer might admit.

Responses to the comments raised by the Reviewer

Comment 1

Figure 1. Images of representative Nissl-stained sections taken from monkeys show weak tissue integrity at the intact left side. It appears at first glance, as if the tissue underwent freezing damages, but is not clear what and how happened. It appears that the injured side contains a relatively intact scar tissue. Please explain and correct these images.

Response to Comment 1

As pointed out by the Reviewer, Nissl-stained sections with weak tissue integrity were included in Figure 1, due to freezing damages and so on as the Reviewer assumed. Therefore, we have replaced the corresponding three sections with the fresh ones.

Comment 2

Suppl. Fig. 1A and B. Why do these images overlap? Please indicate the injured and intact side. Please show the CST at higher magnification (insert high power images as inset, or similar) to visualize better the difference between the groups. The PKCgamma signal is stronger on the injured side than on the intact one. Please explain and amend/replace the figure.

Response to Comment 2

As explained above, we have replaced the data on PKC� immunostaining with those on BDA anterograde tract-tracing, reflecting the presentation manner suggested by the Reviewer (see Supplementary Figure 1A and B). According to this revision, we have also changed and/or added some related descriptions in the corresponding parts of the Materials and methods section (page 5, lines 90–91; page 9, line 203 through page 10, line 213; page 10, lines 216–217; page 11, lines 228–246), the Results section (page 13, lines 285–293), and the Discussion section (pages 16–17, lines 366–377 as well as in the legends of Figure 1 (page 5, lines 111 through page 6, 112) and Supplementary Figure 1 (page 21, line 508 through page 22, line 517).

Comment 3

Suppl. Fig 1C and D. Please insert an overview of spinal cords in lower magnification and indicate the source of higher magnification images (with boxes of sampling).

Response to Comment 3

According to the Reviewer’s advice, we have added spinal sections in lower magnification and indicated the source of higher magnification images (see Supplementary Figure 1C and D).

Comment 4

Furthermore, authors need to indicate the staining in the Fig.1 and Suppl. Fig.1., too.

Response to Comment 4

According to the Reviewer’s suggestion, we have added some descriptions in the legends of Figure 1 (page 6, lines 120–121) and Supplementary Figure 1 (page 22, lines 516–517).

Comment 5

In results section the glia and CST results should be better explored. At present it is just one sentence. Please explicitly discuss them in the discussion section.

Response to Comment 5

As described in Response to Comment 2, we have thoroughly revised the corresponding parts of the Results section (page 13, lines 285–293) and the Discussion section (pages 16–17, lines 366–377).

---

## [Decision Letter · Decision Letter 2]

10 May 2021

An enhanced therapeutic effect of repetitive transcranial magnetic stimulation combined with antibody treatment in a primate model of spinal cord injury

PONE-D-20-28971R2

Dear Dr. Yamanaka,

We’re pleased to inform you that your manuscript has been judged scientifically suitable for publication and will be formally accepted for publication once it meets all outstanding technical requirements.

Kind regards,

Antal Nógrádi, M.D., Ph.D., D.Sc.

Academic Editor

PLOS ONE

Additional Editor Comments (optional):

Reviewers' comments:

Reviewer's Responses to Questions

**Comments to the Author**

1. If the authors have adequately addressed your comments raised in a previous round of review and you feel that this manuscript is now acceptable for publication, you may indicate that here to bypass the “Comments to the Author” section, enter your conflict of interest statement in the “Confidential to Editor” section, and submit your "Accept" recommendation.

Reviewer #1: All comments have been addressed

2. Is the manuscript technically sound, and do the data support the conclusions?

Reviewer #1: Yes

3. Has the statistical analysis been performed appropriately and rigorously? 

Reviewer #1: Yes

4. Have the authors made all data underlying the findings in their manuscript fully available?

Reviewer #1: Yes

5. Is the manuscript presented in an intelligible fashion and written in standard English?

Reviewer #1: Yes

6. Review Comments to the Author

Reviewer #1: The authors have performed all the changes suggested and answered the significant questions. The study can be published in its present form.

7. PLOS authors have the option to publish the peer review history of their article (what does this mean?). If published, this will include your full peer review and any attached files.

Reviewer #1: No

---

## [Editor Report · Acceptance letter]

24 May 2021

PONE-D-20-28971R2 

An enhanced therapeutic effect of repetitive transcranial magnetic stimulation combined with antibody treatment in a primate model of spinal cord injury 

Dear Dr. Yamanaka:

I'm pleased to inform you that your manuscript has been deemed suitable for publication in PLOS ONE. Congratulations! Your manuscript is now with our production department. 

Kind regards, 

on behalf of

Prof. Antal Nógrádi 

Academic Editor

PLOS ONE